Comparative analysis of early ontogeny in Bursatella leachii and Aplysia californica

Vue Zer 1 4 5
Kamel Bishoy S. 1 2 bishoyh@gmail.com
Capo Thomas R. 3
Bardales Ana T. 3
Medina Mónica 1 2 mum55@psu.edu
1 School of Natural Sciences, University of California , Merced, CA , USA
2 Department of Biology, Pennsylvania State University , University Park, PA , USA
3 Rosenstiel School of Marine and Atmospheric Science, Division of Marine Biology and Fisheries, University of Miami , Miami, FL , USA
4 Program in Developmental Biology, Baylor College of Medicine , Houston, TX , USA
5 Department of Genetics, University of Texas M.D. Anderson Cancer Center , Houston, TX , USA
De Baets Kenneth
Electronic publication date: 2014 Dec 11
Publication date: 2014
Volume: 2
Electronic Location ID: e700
Received 2014 Jun 20; Accepted 2014 Nov 25
Copyright: © 2014 Vue et al.
Copyright year: 2014
Copyright holder: Vue et al.
License: This is an open access article distributed under the terms of the Creative Commons Attribution License, which permits unrestricted use, distribution, reproduction and adaptation in any medium and for any purpose provided that it is properly attributed. For attribution, the original author(s), title, publication source (PeerJ) and either DOI or URL of the article must be cited.
License URL: https://creativecommons.org/licenses/by/4.0/

Keywords: Shell loss, Sea hares, Biomineralization, Aquaculture, Larvae

Funding: NSF DEB-0542330 IOS 0926906 Support for this project was provided by NSF DEB-0542330 and IOS 0926906. The funders had no role in study design, data collection and analysis, decision to publish, or preparation of the manuscript.

==============================
Opisthobranch molluscs exhibit fascinating body plans associated with the evolution of shell loss in multiple lineages. Sea hares in particular are interesting because Aplysia californica is a well-studied model organism that offers a large suite of genetic tools. Bursatella leachii is a related tropical sea hare that lacks a shell as an adult and therefore lends itself to comparative analysis with A. californica. We have established an enhanced culturing procedure for B. leachii in husbandry that enabled the study of shell formation and loss in this lineage with respect to A. californica life staging.

Introduction

The Mollusca has been one of the most successful metazoan lineages in exploiting the advantages of the hard, calcified shell (Lowenstam & Weiner, 1989; Weiner & Dove, 2003). Yet there are several molluscan groups that subsequently evolved to have a highly reduced shell, e.g., squid, or have lost it completely, e.g., sea slugs (Kröger, Vinther & Fuchs, 2011; Morton, 1960). Shell reduction or loss has also occurred in euthyneuran gastropods, i.e., marine and terrestrial slugs. Within the sea slugs (Opisthobranchia), shell reduction or loss has occurred in members of the Cephalaspidea, Anaspidea, Sacoglossa, Acochilidiacea, Nudibranchia, and Pleurobranchia among others (Wägele & Klussmann-Kolb, 2005). These events support the notion that shell reduction or loss is not an isolated event and instead has evolved independently many times through parallel evolution (Gosliner, 1985; Gosliner, 1991). Having a slug-like form may be well-suited for a borrowing or swimming lifestyle, which is necessary for streamlining and reducing the weight of the organism (in pelagic forms) (Vermeij, 1993). Shell loss paved the way for extraordinary body plan modifications observed in the different molluscan lineages that underwent this dramatic anatomical change enabling them to occupy new niches such as plastic sequestration from ingested macroalgae for photosymbiosis (Kempf, 1984; Rumpho, Summer & Manhart, 2000) in shell-less gastropods (Vermeij, 2013), camouflage (Rudman, 1981; Rudman & Avern, 1989) and mimicry of unpalatable species (Kerstitch, 1989), swimming to escape danger (Gillette & Jing, 2001; Lawrence & Watson III, 2002), and incorporation of defense mechanisms from their diet organisms as their own, e.g., nematocysts (Conklin & Mariscal, 1977; Greenwood & Mariscal, 1984). Because these adaptations involve anatomical modifications that tend to take place during early development, we consider that differential shell reduction and loss in sea hares provides an excellent opportunity to investigate major transitions in gastropod body plan evolution.

Within the sea hares (Opisthobranchia: Anaspidea), shell reduction or loss has occurred at least twice in adult individuals (Fig. 1) but possibly more times. The order Anaspidea is best known for the work on Aplysia californica as a model system for the study of the cellular basis of behavior (Kandel, 1979) and molecular and genome resources are readily available (Heyland et al., 2011). Transcriptome profiling, combined with whole mount in situ hybridization, has identified differentially expressed genes during shell formation in early developmental stages of A. californica (Heyland et al., 2011) providing a list of candidate genes involved in the process of shell formation that can now be analysed in other anaspidean taxa.

Figure 1 Phylogenetic tree depicting relationships of Anaspidea.

Consensus phylogeny of sea hares (Anaspidea) compiled from Medina & Walsh (2000) and Klussmann-Kolb & Dinapoli (2006). Shell character states are depicted by boxes on terminal nodes.

Although the phylogeny of the Anaspidea is still partly unresolved (summarized in Fig. 1), the monophyly of the group is well supported by several morphological synapomorphies, i.e., reproductive system, defensive glands, radula, gizzard and nervous system (Ghiselin, 1965; Klussmann-Kolb & Dinapoli, 2006; Mikkelsen, 1996; Morton & Holme, 1955), as well as molecular phylogenies (Grande, Templado & Zardoya, 2008; Klussmann-Kolb & Dinapoli, 2006; Medina & Walsh, 2000; Thollesson, 1999). The current understanding of phylogenetic relationships also enables us to map the evolution of shell reduction and loss within the sea hares. While adults of the genus Aplysia exhibit a reduced shell, the genus Bursatella represents the derived character state of crown anaspidean taxa where adults lack a shell altogether. Thus the ragged sea hare, Bursatella leachii exhibits some developmental differences relative to A. californica providing a good comparative system to study shell evolution in this gastropod lineage. Both species undergo two distinct periods of shell growth separated by cessation during the metamorphic process. Following the A. californica life cycle staging (Kriegstein, 1977), characteristic veliger spiral shell growth commences during the encapsulated embryonic phase and continues to the end of the planktotrophic larval phase, stage 6. Growth resumes post metamorphosis at stage 10, when the shell changes from a spiral to a planar shell growth pattern. A. californica has an internalized shell in adulthood, whereas B. leachii undergoes post-metamorphic shell growth followed by shell loss soon after metamorphosis (Paige, 1988).

Aplysia californica is one of a few invertebrate species with long-lived planktotrophic larvae that can be successfully cultured in the lab (Carefoot, 1987; Kriegstein, 1977). Today, after optimized short generation times and developmental inducers, a large number of A. californica can be grown in the laboratory under controlled hatchery conditions. High fecundity and quick growth provide abundant experimental stock of multiple life stages (Capo et al., 2009). With the success of A. californica cultures year-round, having additional hatchery populations of other anaspidean species is an attainable goal given our understanding of the ecology and evolution of related taxa (Carefoot, 1987). Habitat and dietary preferences in B. leachii are now well-known, facilitating animal husbandry. B. leachii lives in tropical subtidal waters (Ramos, Lopez Rocafort & Miller, 1995) feeding on cyanobacterial biofilms found on sandy substrates (Paige, 1988; Ramos, Lopez Rocafort & Miller, 1995).

In this study we report a more detailed description of the B. leachii life cycle than previously available, normalized to the A. californica hatchery culturing procedures currently in place at the National Aplysia Resource facility (Capo et al., 2009). We also report new optimal culture conditions for B. leachii. We conclude by describing the most apparent differences in the developmental program of both species, with emphasis on metamorphic stages during which shell reduction and loss take place with discussion of potential biomineralization proteins involved in shell formation in sea hares.

Methods

Broodstock and eggs/larval rearing

Aplysia californica adults were collected by Santa Barbara Marine Biologicals in 2006. Bursatella leachii adults were collected along the coast of Key Biscayne, Florida during the summer of 2006. All organisms were housed in the flow-through seawater system at the National Aplysia Resource Facility at the University of Miami’s Rosenstiel School of Marine and Atmospheric Science (RSMAS) as previously described (Capo et al., 2009; Capo et al., 2002). The animals were fed a daily ration of the following laboratory-cultured seaweeds: Gracilaria ferox (for A. californica) and a mixture of blue–green algae and epiphytes (for B. leachii). The light cycle of both species was maintained at 12 h light: 12 h dark. The seawater temperature was 13–15 °C for A. californica and 22–26 °C for B. leachii. In year 1 of the study, cultures were maintained at the same temperature (22 °C) but the B. leachii cultures died. In the subsequent trial, parallel cultures were maintained at 22 °C and 25 °C for A. californica and at 25 °C for B. leachii. Mating pairs were monitored throughout the day for active egg-laying. During oviposition, a 10 cm portion of egg strand was collected, rinsed immediately with 0.45 µm filtered seawater, placed in a 2l flask to which Na2 EDTA (0.25 mg/l) was added to bind heavy metals in the natural seawater that may deleteriously affect development (Capo et al., 2002). The eggs and seawater were vigorously aerated until one day prior to hatching in a temperature-controlled incubator at 22 °C and 25 °C for A. californica and 25 °C B. leachii in the last trial of the culturing experiments. Hatching occurred 7–8 days after the eggs were deposited and the cordon (egg strand) was inspected under a dissecting microscope at six days post-oviposition to validate normal and synchronized development of embryos. Strands not meeting these standards were discarded.

The number of larvae/mm of cordon was estimated by cutting three portions of known length, using an ocular micrometer. Each segment was dissolved in 2% sodium hypochlorite and the shells were counted. Day 0 shell length (SL) for both species was determined by measuring 25 individuals from each portion of the cordon using an ocular micrometer at 50X magnification. The appropriate initial larvae density was provided by aseptically cutting the appropriate cordon length, immediately rinsing with 2 µm filtered seawater, and directly transferring it into the larvae culture vessel.

Seawater was collected from Bear Cut, Virginia Key, FL and prepared by prefiltration through a 15 µm glass media filter. The salinity was adjusted to 32 ppt with deionized water, and aerated with chloramphenicol (2.5 mg/l), Na2 EDTA (0.25 mg/l). Eighteen to 24 h later the seawater was vacuum filtered through a 2 µm prefilter (Millipore AP2504700) (Kriegstein, Castellucci & Kandel, 1974; Nadeau et al., 1989). The desired concentration of microalgae and estimated length of egg mass were added to filtered seawater in 2 L roller bottles (Corning). The vessel was sealed with Parafilm® and plastic wrap to eliminate the air-water interface (Capo, Perritt & Paige, 1987; Paige, 1986; Tamse, Kuzirian & Capo, 1990). The cultures were incubated on a continuously rotating (1 rpm) roller bottle apparatus (Wheaton), with a 24 h fluorescent light regime (∼0.001 µE/cm2/s) at a constant temperature of 22 °C (Kriegstein, Castellucci & Kandel, 1974; Nadeau et al., 1989; Tamse, Kuzirian & Capo, 1990). Roller apparatus positions were randomly assigned to each culture vessel and remained fixed throughout the experiment.

After hatching, larvae were measured and the culture media was changed every 7 days. The larvae were collected on a 74 µm mesh screen, rinsed with filtered seawater (FSW) and transferred to a sterile crystallizing dish. An iodine-based surfactant (Betadine Surgical Scrub) was added to resuspend any larvae entrapped by the air-water interface. Larvae were treated with 1.25 ml of a solution of 2.5 mg/ml Poly (vinylpyrrolidone)–Iodine complex (Sigma) and 2.0 mg/ml pH 8.3 fish-grade Trizma (Sigma) solution for 5 min to inhibit bacterial growth. This treatment also acted to suppress larval swimming behavior and provided a non-lethal method to facilitate shell length measurements. Weekly SL of 25 larvae was measured and the larval stage for both A. californica and B. leachii was determined through Kriegstein’s staging scheme for A. californica (Kriegstein, 1977). Once the exposure period ended, the iodine concentration was reduced by the incremental addition of a 0.4% sodium thiosulfate solution to the treatment bath until the characteristic iodine color had disappeared. The larvae were rinsed in FSW and transferred to a clean, acid-washed roller bottle with FSW containing the appropriate amount of microalgae and sealed (250 × 103 cells/ml, Isochrysis sp.—CCMP1324). The bottles were then returned to the previously assigned roller bottle apparatus and position.

For imaging of each stage of B. leachii, the larvae were placed in filtered seawater (0.22 µm) containing 340 mM of magnesium chloride. Once animals were narcotized, photographs were taken with an Olympus BX51 microscope or a Leica MZ16F stereoscope. Scanning Electron Microscopy was performed on a limited numbers of larval shells from both species.

Results

Post-hatching larval development and shell growth

The life cycle staging of B. leachii mentioned here is equivalent to the staging scheme that was described for Aplysia californica (Kriegstein, 1977) and currently in use at the University of Miami’s Aplysia hatchery (Rosenstiel School, 2012). Stage 1 is characterized by a newly hatched veliger containing a Type 1 shell (Thompson, 1961). In B. leachii, Stage 1 larvae have a maximum shell diameter of 141.1 ± 6.9 µm (N = 25) and the veliger’s shell grows rapidly—an average of 21 µm per day (Fig. 2). Stage 2, defined by the appearance of the eyes, and is reached within 4 days post-hatching. By day 5, the shell length is 264.6 ± 13.9 µm (N = 25) with the presence of 1.5 whorls. After 6 days post-hatching, the larval heart appears (Stage 3). By day 7, the maximum shell size (Stage 4) is reached at 284.2 ± 19.0 µm (N = 25) (Supplemental Information 1). Almost at the same time the foot expands to form a well-developed propodium (Stage 5). On day 9, the larvae reach competency and settle (Stage 6) when exposed to a substratum. A morphological pigmented spot on the shell, similar to A. californica (Kriegstein, 1977), is also present in B. leachii. Paige (1988) and Paige (1986) failed to observe and report pre-metamorphic pigmentation most likely due to the use of artificial seawater. Pigmentation is a clear indicator of competency to metamorphose, and can be reached as early as 9 days post-hatching. A. californica larvae grown at 22 °C and 25 °C showed that there was no difference in growth. A two-way repeated measures ANOVA reflects that there was no difference in the size of A. californica grown at 22 °C vs. 25 °C (Supplemental Information 1 and Supplemental Information 2). In 2006, total mean shell length (n = 25) for A. californica grown at 22 °C averaged 134.6 µm (s = 3.7 µm) for Stage 1, 227.6 µm (s = 15.0 µm) for Stage 2, 337.7 µm (s = 20.8 µm) for Stage 3 and 392.8 µm (s = 10.0 µm) for Stage 5. Total mean shell length (n = 25) for A. californica grown at 25 °C averaged 134.6 µm (s = 3.7 µm) for Stage 1, 236.1 µm (s = 18.6 µm) for Stage 2, 360.6 (s = 36.9 µm) for Stage 3 and 392.3 µm (s = 18.9 µm) for Stage 5 (Supplemental Information 2).

Figure 2 Larval and juvenile growth of Bursatella leachii and Aplysia californica in laboratory settings.

Veliger shell length of A. californica and B. leachii larvae grown at 25 °C in 2006. Shell length was measured weekly from day of hatching until 80% competency; error bars represent ±1 standard deviation. Arrow indicates timing of competency: 9 days post-hatching in B. leachii and 22 days post-hatching in A. californica. Previous attempts to culture B. leachii larvae at 22 °C were unsuccessful (not shown).

Metamorphic larvae development of Bursatella leachii

Metamorphic development and post-larval development of Bursatella leachii is similar to other previously described sea hares (Kriegstein, 1977; Paige, 1988; Switzer-Dunlap, 1978; Switzer-Dunlap & Hadfield, 1977). At Stage 5, the propodium forms an essential structure needed for settlement and crawling after settlement. At Stage 6 (Fig. 3A), metamorphic competence occurs, along with the appearance of other morphological traits, such as a pigmented spot on the right side of the perivisceral membrane underneath the shell (Kriegstein, 1977). Once the larva has settled, in the presence of an environmental cue (Heyland, 2006; Paige, 1988), it will attach itself permanently and shed its velar lobes (Stage 7) (Fig. 3B). The metamorphic transition occurs when the two halves of the velar lobe rudiments fuse together and the larval heart stops beating, which is also an indicator of Stage 8. Post-metamorphic shell growth in both A. californica and B. leachii (Stage 9) is characterized by an elongation of the larval shell (Fig. 3C). Stage 10 is reached when the shell reaches its maximum size and flattens prior to being discarded (Fig. 3D). The shell is discarded at Stage 11, when the juvenile begins to show adult characteristics. Figure 3E shows a late Stage 11 juvenile, approximately 2 mm long, after discarding its shell. The juvenile takes on adult characteristics, such as the appearance of small bumps all over the body and rudiments of the fleshy villae. The rhinophores are elongated and tubular and the oral tentacles expand laterally. The body is pigmented with large, white granular patches. At Stage 12, Bursatella leachii (Fig. 3F) is approximately 8 mm long. The villae cover the entire body, multiply and become branched later in adulthood. Shell development is similar in early embryonic stages but diverges as juvenile development takes place leading to shell loss in B. leachii. We examined by SEM both whole shells and cross-sections of larval shells (Supplemental Information 3). Despite some noticeable similarities between the two species, unfortunately due to the small size of the larval shells, we either did not have enough replicates per stage or missed stages altogether to raise clear conclusions about larval shell shape and internal structure.

Figure 3 Metamorphic development of Bursatella leachii.

Metamorphic competence of the veliger larvae (stage 6, A) correlates with many morphological characteristics (i.e., red spots, propodium, etc.). Once settled, the larvae will attach and shed their velar lobes, becoming benthic (stage 7, B). Stage 8 (not shown) marks the end of metamorphosis, characterized by the fusion of the two halves of the velum lobe and the loss of the larval heartbeat. Stages 9–10 marks the development of specific morphological structures of juveniles, such as the elongation of the juvenile or post metamorphic shell (stage 9, C; stage 10, D). Adult characteristics, such as the complete shedding of the shell, rhinophores, villae and oral tentacles, will start to appear in late stage 11 (E) and the adult (F). VL, Velar Lobe; Sh, Shell; Sp, Spot; M, Mouth; F, Foot; E, Eye; Pp, Propodium; R, Rhinophores; OT, Oral Tentacles; Vi, Villae. Scale bar in A: 100 µm, in B: 67 µm, in C: 108 µm, in D: 134 µm, in E: 254 µm, in F: 1mm.

Discussion

The life cycle of Bursatella leachii was characterized in reference to the well-known A. californica life cycle. Having access to the complete life cycle of a second anaspidean species enables comparative developmental studies within the sea hare clade. In the present study we describe the life cycle of B. leachii in the context of the development of the larval shell and its subsequent loss in the post-metamorphic stages.

Bursatella leachii development

The embryonic development of Bursatella leachii has been described previously (Bebbington, 1969; Paige, 1988) and thus will not be further discussed here. The larval developmental sequence of B. leachii is similar to other sea hares (Switzer-Dunlap, 1978)—a hatched veliger with a hyperstrophically coiled shell, a reddish tint, and bilobed velum. B. leachii larvae differ both in size and growth rate relative to A. californica, being both larger (approximately 10 µm) and faster growing, though the larval development follows the staging sequence previously devised in the literature (Kriegstein, 1977; Paige, 1988). Similar to Kriegstein (1977), our study demonstrated the presence of one prominent Stage 6 pigmentation spot in B. leachii.

Initial stages of post-metamorphic development of sea hares with a planktonic larval form are also similar, Table 1 summarizes the larval development of B. leachii (Paige, 1988) relative to A. californica (Capo et al., 2009; Kriegstein, 1977). Recent advances in larval culture techniques provide the tools for life cycle comparisons. The need for readily available developmental stages is important for experimental developmental biology studies such as metamorphic transitions. In the particular case of sea hares, hatchery populations provide an ideal supply of samples for the study of larval shell loss.

Table 1 Comparison of developmental schedules of Aplysia californica and Bursatella leachii.

Comparison of morphological development schedules of A. californica larvae as reported by Kriegstein (1977) compared to Capo et al. (2009) and comparison of B. leachii larvae as reported by Paige (1988) compared to the present study. Values are the number of days post-hatching until the specified developmental stage was observed.

Stage	Description	Bursatella a	Bursatella e	Aplysia b	Aplysia c	
2	Eyes	6	4	14	7	
3	Larval heart	12	6d	21	14d	
4	Maximum shell size	15	7	28	17	
5	Propodium	17	7	30	19	
6	Competency	19	9	32	22	
6	Red spots	None	1 large spot	Present	Present	
7	Metamorphosis	20	12	34	24	
Notes.

a Paige, 1988.

b Kriegstein, 1977.

c Capo et al., 2009.

d 50 beats/minute not taken into consideration.

e Present study.

Differences after metamorphosis occur at Day 40 during Stage 9 when A. californica juveniles acquire pink pigmentation due to the red algal diet, while B. leachii juveniles become white with dark bands on the head (Paige, 1988). Despite this post-metamorphic physical difference, their developmental programs remain highly similar to each other up until this point. A major difference in B. leachii post-metamorphic development happens at Stage 11 when the shell is discarded. At this stage in A. californica, the shell becomes overgrown by folds of the mantle, causing the shell to be internalized. Given that both species follow a similar developmental program through metamorphosis, it seems quite plausible that the underlying mechanism of larval shell formation is also quite similar, only differing during settlement/post-metamorphosis. We conclude that larval shell formation appears to be homologous in these two species, which makes this process amenable to comparisons such as the examination of spatio-temporal gene expression of genes involved in the formation of the shell in both species. It seems plausible that the evolution of shell loss is the consequence of modifications to the regulatory machinery of shell formation genes, as most molluscs have the ability to make shells at least in the embryonic stages.

Shell development in Anaspidea

Shell building in molluscs is on the cellular level characterized by modifications to the extracellular matrix (ECM) that create an environment conducive to crystal deposition in the extrapallial space. Analysis of the shell “secretome,” during calcification in the abalone, Haliotis asinina by Jackson et al. (2006) yielded a significant number of transcripts. A direct comparison of the transcriptomes of nacre-forming cells from H. asinina (gastropod) and Pinctata margaritifera (bivalve) led to the conclusion that there are dramatic differences in the gene sets used to build the nacreous layer of the shell (Jackson et al., 2010). These differences also extend within the Gastropoda (H. asinina vs. Lottia gigantea). A comparison of a single biomineralizing gene family (shermatin) across three species of Pinctata, suggested that secreted proteins with repetitive low-complexity domains (RLCDs) are an important feature in molluscan evolution but are the consequence of evolutionary convergence (Jackson et al., 2010) thus supporting the notion that the molluscan shell-secretome is rapidly evolving (Jackson et al., 2006). The rapid evolution scenario complicates questions of functional homology across species as many of the biomineralization proteins provide multiple other functions such as immune response (Sarashina et al., 2006). Work on early developmental stages where the shell is starting to form is of relevance to this study. Heyland et al. (2011) detected 196 different transcripts that appear to be related to biomineralization in a developmental transcriptome time course in Aplysia californica. These 196 transcripts were present during the whole course of development and although not unique to the veliger stage, they were slightly overexpressed during the veliger/trochophore stage and several are well known biomineralization proteins reported for other molluscs such N66, Perlucin, Pearlin and Nacerin (Heyland et al., 2011). Reported gene expression throughout the entire course of development hints at the fact that larval shell development in Aplysia is primarily executed via regulatory mechanisms. The majority of the detected transcripts lack annotation highlighting the importance of functional studies for the discovery of new biomineralization-related proteins. The ability to transfer this information into B. leachii would enable us to test multiple hypothesis about how conserved are the mechanisms of shell building during early development in sea hares, a crucial step in increasing our understanding of the fascinating phenomenon of biomineralization and evolution of shell loss in opisthobranchs.

B. leachii husbandry

We present an improved strategy to culture B. leachii in larger numbers than previously reported. We attempted to rear both species under similar conditions but A. californica is a temperate species from the Western North America, where coastal upwelling is prevalent and water temperatures low relative to tropical waters where B. leachii is common. Therefore we decided to use a slightly higher temperature (25 °C) for the second year the cultures were established in the lab. The primary goal of this study was to produce individuals from comparable stages, however, despite small sample sizes and limited controls, our efforts have lead to an improved culturing method for B. leachii with larger larval yields than previously reported (Paige, 1988).

Conclusion

We have established a reliable culturing technique for B. leachii that makes this species amenable to experimentation at all developmental stages (Capo et al., 2009). Transcriptome data and whole mount in situ hybridization available for A. californica (Heyland, 2006) have enabled developmental genetics research (Heyland et al., 2011) in anaspideans. While comparative studies of biomineralization genes in sea hares are in their infancy, with developmental homology clearly established and an improved cultivation protocol, we are primed to shed light on how the genetic toolkit that controls shell formation and subsequent reduction or loss differs between A. californica and B. leachii.

Supplemental Information

Supplemental Information 1 Comparison of larval and juvenile growth of Bursatella leachii and Aplysia californica in laboratory settings

Veliger shell length of A. californica and B. leachii larvae grown at 25 °C and A. californica larvae grown at 22 °C and 25 °C in 2006. Shell length was measured weekly from day of hatching until 80% competency, error bars represent ±1 standard deviation. Values are the number of days post-hatching until the specified developmental stage was observed.

Click here for additional data file.

Supplemental Information 2 Comparison of larval and juvenile growth of Aplysia californicain laboratory settings at 22 °C and 25 °C

Mean shell length (mean ± standard deviation (Stdev)) of Aplysia californica larvae grown at 22 °C and 25 °C for stage 1, 2, 3 and 5 in 2006. Mean shell length at 22 °C for each stage n = 25 shells; Mean shell length at 25 °C for each stage n = 25. Two-way Repeated Measures ANOVA, 18 df, p < 0.0001.

Click here for additional data file.

Supplemental Information 3 SEM of Aplysia californica and Bursatella leachii larval shells

Whole shell of Stage 6 veligers of Aplysia californica (A) and Bursatella leachii (C) and cross sections of A. californica (B) and B. leachii (D)

Click here for additional data file.

We thank Phillip Gillette for his efforts in culturing B. leechii; Benoît Dayrat for help with Fig. 1; Alice Hudder for training. We also acknowledge the support of Michael R. Dunlap and the Imaging and Microscopy Facility (IMF) at the University of California, Merced. This manuscript was prepared by Zer Vue in partial fulfilment of requirements for the master’s program in Quantitative Systems Biology at UC Merced. We thank Chris Voolstra, Michael DeSalvo and Shini Sunagawa for providing feedback on an earlier version of this manuscript.

Additional Information and Declarations

Competing Interests

Author Contributions

Mónica Medina is an Academic Editor for PeerJ.

Zer Vue, Bishoy S. Kamel, Thomas R. Capo, Ana T. Bardales and Mónica Medina conceived and designed the experiments, performed the experiments, analyzed the data, contributed reagents/materials/analysis tools, wrote the paper, prepared figures and/or tables, reviewed drafts of the paper.

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
