# Peer review of "Comparative analysis of early ontogeny in Bursatella leachii and Aplysia californica"

_PeerJ, doi:10.7717/peerj.700_

## Round 0.1 · original submission · Major Revisions

You provide interesting new data on the development and growth rates of Bursatella leachii which might be important to properly understand the evolution of shell reduction and loss in gastropods. It is important to study taxa other than the traditionally studied model organisms (e.g., Aplysia). However, the reviewers have raised some critical points, which justify major revisions at best (one of the referees even recommended rejection). Due to differing opinions of the reviewers, I wanted to give you a chance to address these points and let the manuscript go into another round of reviews. The manuscript title, introduction and needs to be more coherent and better structured to make clear what is studied. The title eludes to the study of shell development and the introduction eludes to establish a comparative scheme for the reduction or loss of shells in B. leachii, but mostly normal development processes of the post-embryonic specimens and some photographs with the SEM of post-metamorphic stages are provided (see reviewer 2). The following points need to be addressed:

Mating and raring temperatures: why did you choose different temperatures for mating of the adults and raring the hatchlings? The text also mentions a different temperature (22°C) than the figures (25°C). Temperature might influence growth rates, particularly A. californica which might prefer lower temperature (at least for mating) than B. leachii.

Growth rates: you mention growth rates as the main difference between the two taxa, but no clear explanation is given in the text why this could be the case (see also comments of reviewer 2). Couldn´t this be related with temperatures (you raised the adults in different adults in different temperature, but raised the hatchlings at 22°C); furthermore as remarked by reviewer 1 (one taxon is considerably larger than the other; so larger growth rates would be logical).

Shell development during early ontogeny: Only pictures of complete shells of stage 6 are provided, (see also comments by reviewer 1. 2), but it would be critical to also provide pictures/drawings of earlier and later stages discussed in the text. I agree with both reviewers that more pictures of shells need to be provided of different stages and more shells of stage 6 (see reviewer 1) to be able to demonstrate and discuss the similarities/differences in the shell development; Please integrate pictures and observations of earlier stages between these taxa back to back to be able to create a comparative scheme for the reduction or loss of shells in B. leachii in relationship with A. californica.
Characterization of shell growth: Reviewer 2 correctly points out that to characterize the specific mode of the shell growth in Bursatella, the figure 2 would be the most important, particularly day 5 of post-hatchling. It would be interesting to have information on what happening during this event (e.g., cell death, inhibition of shell secretion, or loss of excretory cells), which could be obtained by investigating more cellular events by using the useful comparative models.

Citations: On various occasions, crucial statements are made without providing adequate references (e.g., concerning the Cambrian explosion, advantage of a slug-like form for streamlining and reducing weight in the introduction, predictions are made about gene expressions and cis-regulatory regions on p. 12, etc.). These are discussed in more detail in my remarks.

Novelty: It is not always clear what is new in your study (compare reviewer 1); please state this more clearly.

Evolutionary origin: you do not go into the fossil record of Bursatella and their large group, which is probably rather poor (e.g., Medina et al. 2001 discussed this for Aplysia). It would be interested to mention when the first forms with reduced shell probably first appeared based on molecular clock and the fossil record as you already mention the evolutionary radiation of shelled mollusks earlier in the introduction. For Aplysia this was discussed for example by Medina and Walsh (2000) and Medina et al. (2001)

I would like you to address the following points that I noted in the manuscript in addition to the ones raised by the reviewers:

p. 1, line 1-2 (title): the title eludes to the study of “shell development in sea hares”; maybe drop this part as you mainly do a classical study of normal early ontogenetic development (compare reviewer 2)
p. 2, lines 1-15: a large part of the abstract is a repetition of what´s written in the introduction and does not directly describe you main research question or the research you did for this paper; please rephrase
p. 2, line 8-9: you mention that you compare both species, but you provide new data on Bursatella leachii, which you compare with previously data of Aplysia; this should be changed to reflect this
p. 2, line 17-20: you mention the „ explosion“ of hard-shelled animal taxa in the fossil record without providing adequate references to this Cambrian explosion (e.g., Erwin et al. 2011); Furthermore, its age and the appearance of shelled organisms is much better known than “about 500 Ma” (Murdock and Donoghue 2011; Kouchinsky et al. 2012). Please cite these references in this context (there are listed at the end of this list).
p. 4, line 15-17: “In particular, the mechanisms of shell development in these two species appear to be similar in early embryonic stages but diverge as juvenile development takes place”. It sounds like you already know the outcome of the study, before you did it. You need to rephrase this paragraph or keep it for the results and just mentioned that previous studies have focused on Aplysia.
p. 4, line 6-10: Is there one of these theories which is more supported by morphology/ontogenetic development? Which one do you prefer. Please call to figure 1 in this context.
p. 6, line 18-20: So A. californica does better at lower temperature (13-15°C) and B. leachii at higher temperatures (22-26 °C) ? Why did you choose these temperatures? Why is one written as 14 +/- 1°C and the other as a temperature range ?
p. 6, line 24-25: please change to “at 22°C for both A. California and B. leachii”. Why did you choose this temperature ? As also noted by one of the reviewers you mention 22°C in the text, but in figure 2 you refer 25°C. Please verify and correct.
p. 8, line 17-18: you need to provide pictures of the intact and broken pieces from stage 6 larvae (compare reviewer 1)
p. 9, line 6-19; please provide figures (pictures or drawings of specimens and particularly shells) of all these stages (1-9; not just stage 6) before and after metamorphosis of both taxa; Please also add the stage where the measurements are from to Figure 2 for clarity.
p. 12, line 6-11: several statements and predictions are made about gene expressions and cis-regulatory regions without any references; please add some citations to corroborate this
p. 12, line 22: why do you just provide images of shells of stage 6 and not of other stages; at least figures or picture should be possible as you do report the measurements; please show pictures of these
p. 12, line 24: “morphologically similar” is quite general; be more specific (form?, structure, etc.)
p. 12, line 25: “three distinct layers”; you characterize these layers very superficially; what about the microstructure and organic material? These can be analyzed with element analysis or quantify microstructure (orientation crystals, etc.)
p. 13, line 5-12: this paragraph is very speculative and your data has no bearing on this because you did not analyze the shell composition or organic layers of your specimens.
Figure 2: Shouldn´t this be 22°C as mentioned in the text; Why did you choose this temperature as it might be more ideal for one taxon than for the other and might influence growth rates. You need to add which stages these measurements represent. How many specimens were measured in each sampling (please specify). Please give the original measurements in a supplementary table which can be useful for additional studies. Why did you only analyze 4 points in Aplysia and 5 in Bursatella; this needs to be explained in the caption and/or text.
Figure 3: Why are there no picture/figures of earlier stages or later stages as you obviously measured them ?
Figure 4: Why are there no picture/figures of earlier stages or later stages as you obviously measured them ?
Table 1: Why are there so large differences between different schemes for the same taxon (Paige 1988 vs yours for Bursatella; Kriegstein 1977 vs Capo et al. 2009 for Aplysia) ? Please explain Are growth rates dependent on environmental conditions (temperature) or phylogeny (different species) ? Why are 50 beats/minute not taken into consideration? Please explain.

Suggested references:

Erwin DH, Laflamme M, Tweedt SM, Sperling EA, Pisani D, and Peterson KJ. 2011. The Cambrian Conundrum: Early Divergence and Later Ecological Success in the Early History of Animals. Science 334:1091-1097.
Kouchinsky A, Bengtson S, Runnegar B, Skovsted C, Steiner M, and Vendrasco M. 2012. Chronology of early Cambrian biomineralization. Geological Magazine 149:221-251.
Medina M, Collins TM, and Walsh PJ. 2001. mtDNA ribosomal gene phylogeny of sea hares in the genus Aplysia (Gastropoda, Opisthobranchia, Anaspidea): implications for comparative neurobiology. Systematic Biology 50:676-688.
Murdock DJE, and Donoghue PCJ. 2011. Evolutionary Origins of Animal Skeletal Biomineralization. Cells Tissues Organs 194:98-102.
.

Reviewer 1 ·

Basic reporting

The manuscript is generally well-written and appears to conform to an appropriate template. I have some concern about how much really new information is included.
In sections below I have suggested some improvements that can be made.

Experimental design

In the Methods, the authors should include the temperature at which larvae were raised (probably 22 degrees C; the same as for the egg masses) but Figure 2 compares growth rates of A. californica and B. leachii at 25 degrees C. Temperature is an important factor in larval growth. Also, "appropriate amount of microalgae" is insufficient to describe larval food; the phytoplankton species used and estimated concentration of food needs to be included should someone want to replicate the work.

Validity of the findings

I'm not sure how much new information is included in this manuscript, but the findings reported are valid.
Figure 1 title should read "Phylogenetic tree depicting relationships of Anaspidea." The tree as shown indicates that Stylocheilus has a shell and it should be no shell. Stylocheilus loses its shell soon after metamorphosis just like Bursatella. This is a fairly significant error.
Figure 2 - I think this is only larval growth; does it include juvenile growth? If not, change the caption. Remove "trajectories" and "were" out of page 15, line 4. Clarify if growth rates shown are from 22 or 25 degrees C and be consistent with information in Methods. Was 80% competency at 15 days for B. leachii and 21 days for A. californica in Figure 2? How does this compare with the information in Table 1 of 9 days and 22 days to competency, respectively? Addition of an arrow to indicate age of competency in Figure 2 might clarify this point.
On Figure 3 C and D, use "Sh" instead of "S" to indicate Shell. Page 16, Lines 5 and 6, change "velum" to "velar".
Authors seem to key on B. leachii larvae being larger and faster growing, but don't mention that A. californica is considerably larger than B. leachii at competence. There is a considerable difference in shell size at hatching and competence in larvae of Hawaiian aplysiids that have been cultured.

Additional comments

It would have been interesting to see SEM images of the B. leachii shells lost after metamorphosis and the internalized shells of A. californica at a similar developmental stage since your paper is about shell development.
Minor editorial edits follow:
Pg. 3, Line 12, remove one of the periods after (Vermeij 2013).
Pg. 3, Line 22, remove "W.H." before Watson
Pg 10, Line 10, change to "long, after discarding its shell."
Pg 12, Line 25, lower case "an outer (periostracum) layer;"
Pg 13, Line 10, change to "...B. leachii, the shell is lost post-metamorphosis"
Pg 21, Lines 42-45, why is title in all caps?

Reviewer 2 ·

Basic reporting

The authors compared the normal developmental processes of two sea hares
to obtain insights into the developmental transitions in the shell morphology with different degree of shell reduction. The reviewer understands that the
study of postembryonic development is not easy and establishment of
rearing method of the planktonic larvae is important point, however the
present manuscript unfortunately fails to meet standards of typical journal and it is not well arranged without focus along the main title and story. Also, the scientific validity and their suitability to join the scholarly literature would not be standard.

In the abstract, the authors claimed that the most important finding
of this study was identification of the faster growing rate and size of
B. leachii compared to Aplysia. However, it is not remain uncertain that
why the differential growth rate could characterize the shell-loss or
shell reduced gastropod species. The reviewer could not understand the
possible mechanisms.

Experimental design

The authors described the normal postembryonic development and compared the shell growth in two species, and they found two distinct rates of shell growth.

To characterize the specific mode of the shell growth in Bursatella,
the figure 2 would be the most important, particularly day 5 of
post-hatchling. In the figure, the mean standard length indicates the
termination of growth. But it is not clear what happened during such
critical event: cell death, inhibition of shell secretion, or loss of
excretory cells? The authors could investigate more cellular events by
using the useful comparative models.

Validity of the findings

The purpose of the current study is totally not clear, therefore the validity of the findings tends to be weak. The title says that this study is on the shell development. The introduction says that this study is important to establish comparative scheme for the
reduction or loss of shells in Bursatella leachii compared to a well-examined model species Aplysia. However, the results only showed the normal development processes of the post-embryonic specimens and some photographs with a scanning electron microscope without identification of any specific differences.

Additional comments

The authors compared the normal developmental processes of two sea hares
to obtain insights into the evolutionary and developmental transitions in the shell morphology with different degree of shell reduction. The reviewer understand that the
study of postembryonic development is not easy and establishment of
rearing method of the planktonic larvae is important point, however the
present manuscript is not well arranged without the focus along the main story. The normal development was mainly described and authors found different rate of shell growth, but more detailed histological and cellular level analyses would be required to resolve the main questions and mechanisms on the differential modes of shell reduction in two species.

To characterize the specific mode of the shell growth in Bursatella, the figure 2 would be the most important, particularly day 5 of post-hatchling. In the figure, the mean standard length indicates the termination of growth. But it is not clear what happened during such
critical events: cell death, inhibition of shell secretion, or loss of excretory cells? The authors could investigate more cellular events by using the useful comparative models.

---

## Round 0.2 · Minor Revisions

Thank you for following the comments of the reviewers and myself. I appreciate the restructuring of the paper, so that the manuscript (including interpretations) better reflect the novel data you compiled. From my point of view the paper is as good as accepted (no further reviews are necessary, apart from mine). Your paper will be accepted pending upon the resolution of some minor formatting problems and addition of an additional reference. These minor things refrain me from accepting the current version, because no additional changes can be made to the manuscript after acceptance. I apologize for the inconvenience and thank you for your understanding.

Please change/update:

line 15: please cite Kröger et al. 2011 in the context of reduction of the shell as they extensively review the evolution of cephalopods including the reduction of the shell. (Kröger B, Vinther J, and Fuchs D. 2011. Cephalopod origin and evolution: A congruent picture emerging from fossils, development and molecules. Bioessays 33:602-613.)

line 27: "photsymbiosis" should be replaced with "photosymbiosis"

line 33: please complete/rewrite as something is missing after "during early"

line 90: "The" should be written without capitals as "the"

line 91: "22°C and 25°C": please make this more that parallel cultures were held at two temperatures

line 186: One could wonder why these "data" are "not shown". It advice to write at the end of line 189 that you refrain from adding these data as it is incomplete or something instead of writing this "data not shown" or add it to the supplementary material so readers can decide for themselves.

line 228: please remove the second "is" as it is a duplication which is unnecessary and out of place

line 258-259: The sentence "We ... reported" is repeated at the end of this paragraph ", we have developed ...". Please delete the first one (or rewrite it to avoid saying two times the same thing which is unnecessary in this context".

line 265: please verify the formatting of "Paige, 1988" as it has a different font size or even type

line 283: please verify if all references are listed

line 284: The reference "2012. National Resource Facility for Aplysia. Vol. 2012" is incomplete. Maybe just add it in text and delete from the reference list as it might not belong here if it is written correctly.

caption table 1: please add a space in between "of" and "Aplysia" as well as "californica", "and" and "Bursatella"

figure 1: please add that you mean the "adult shell" with shell reduced and no shell" by adding reduced adult shell or no adult shell"

---

## Round 0.3 · Minor Revisions

Thank you for integrating the requested changes. I only noticed one more small error, which needs to be corrected before i can sent it off into production. Thanks for taking up my advice of citing Kröger et al. 2011. It is however cited two times. The second time, it is in the wrong context: "Within the sea slugs (Opisthobranchia), shell reduction or loss has occurred in members of the Cephalaspidea, Anaspidea, Sacoglossa, Acochilidiacea, Nudibranchia, and Pleurobranchia among others (Kröger et al. 2011; Wagele & Klussmann-Kolb 2005)." Please remove this second citation as to my knowledge Kröger et al. 2011 only discuss cephalopods. This minor points refrains me from accepting the current version, because no additional changes can be made to the manuscript after acceptance. I apologize for the inconvenience and thank you for your understanding one more time.

---

## Round 0.4 · accepted · Accept

Thanks for implementing this final change. Your article is now officially accepted!